# Design Parameters of a Direct Contact Membrane Distillation and a Case Study of Its Applicability to Low-Grade Waste Energy

**DOI:** 10.3390/membranes12121279

**Published:** 2022-12-17

**Authors:** Bitaw Nigatu Tewodros, Dae Ryook Yang, Kiho Park

**Affiliations:** 1Department of Chemical Engineering, University of Gondar, Maraki, Gondar 196, Ethiopia; 2Department of Chemical and Biological Engineering, Korea University, 145 Anam-ro, Seongbuk-gu, Seoul 02841, Republic of Korea; 3School of Chemical Engineering, Chonnam National University, 77 Yongbong-ro, Buk-gu, Gwangju 61186, Republic of Korea

**Keywords:** desalination, membrane distillation, DCMD, low-grade energy, hybrid

## Abstract

In the design of membrane distillation systems, the effect of different heat transfer coefficient models on the transmembrane flux seems to have been overlooked thus far. Interestingly, the range of discrepancy in the results of the transmembrane flux is wide, especially in the laminar flow region, where MD is often operated. This can be inferred by studying the design and parameters of the direct contact membrane distillation system. In this study, the physical and physiochemical properties that affect the design of MD are comprehensively reviewed, and based on the reviewed parameters, an MD design algorithm is developed. In addition, a cost analysis of the designed MD process for low-grade-energy fluids is conducted. As a result, a total unit product cost of USD 1.59/m^3^, 2.69/m^3^, and 15.36/m^3^ are obtained for the feed velocities of 0.25, 1 and 2.5 m/s, respectively. Among the design parameters, the membrane thickness and velocity are found to be the most influential.

## 1. Introduction

Membrane distillation (MD) has been under development in different parts of the world simultaneously, albeit independently. In the early days of this development, MD, owing to its liquid-repulsing-gas-allowing membranes and other related volatile-material-separating processes, was referred to as the “gas membrane”, and the initial concept was presented by Watanabe and Miyauchi in 1976 in Japan [1,2]. Prior to that, in Europe, A. Van Haute et al. attempted to establish a membrane system for permeating water vapor relative to temperature by circulating air above the membrane [3]. However, before the concept of gas membrane or membrane distillation was developed, the first liquid–gas separation system using silicon membrane had been patented in the US in 1963. Thereafter, the use of hydrophobic membranes for the thermal separation of liquid mixtures started to gain popularity in the early 1980s [4,5]. M.E. Findley et al. seemingly published the first study on vaporization and its economic analysis by using paper, glass fiber, and diatomaceous-earth-containing porous membranes, and some of their results led to patents [6]. Nevertheless, the study of MD was popularized in the 1980s after the development of polytetrafluoroethylene (PTFE) membranes. G.C. Sarti et al. studied the use of PTFE composite membranes in a process for separating water from sodium chloride and ammonium chloride solutions [7]. Subsequently, a new patented modular process that utilized hydro energy for seawater desalination was developed in Sweden, with an energy consumption of 1.25 kWh/m^3^ [8]. In the same year, an apparatus and method for thermal MD that utilized the composite membrane was patented in the US [9].

Although this system is not yet ready for desalinating raw seawater, it has been used in processes where waste heat is available. Moreover, it has been hybridized with other membranes and powered using solar energy. Fundamentally, the MD process involves the mass transfer of water vapor through a microporous membrane, coupled with the heat transfer across the membrane and through the boundary layers adjacent to the membranes surface. In MD, the hydrostatic pressure gradient is negligible. Therefore, the driving force of the process is the partial pressure difference across the membrane. In other words, it is the first membrane process where the flux is driven by a thermal gradient between the feed and the permeate streams. The other membranes used in MD are hydrophobic microporous membranes that preventing the transport of liquid across the membrane [5,10,11,12,13,14].

There exist four configurations of MD based on the contacting fluid between the membrane face and the permeate stream: Direct Contact Membrane Distillation (DCMD), Air Gap Membrane Distillation (AGMD), Vacuum Membrane Distillation (VMD), and Sweep Gas Membrane Distillation [10,12,15]. In this study, the focus is on DCMD, owing to its simplicity [16]. Since its development, DCMD has been the most widely used MD configuration [17]. Moreover, it appears to be the best configuration for applications in which the major feed component is water, such as desalination and the concentration of aqueous solutions. DCMD has been successfully applied in areas where treatments at lower temperatures are preferred over those at higher temperatures to safeguard product quality [18].

Apart from being proposed as an alternative membrane technology, MD has not been widely commercialized yet. At the pilot scale, MD is extremely important for producing ultra-distilled water meant for experimental purposes. Moreover, MD is important for high-temperature-sensitive food processing. Spiral wound and hollow fiber modules are commonly used in experimental as well as pilot plant studies. However, because of the high surface area to volume ratio and low recovery rate of MD processes, hollow fiber modules are preferred over spiral wound modules [11,19].

One of the bottlenecks in MD for seawater desalination is its low rate of mass production with respect to the amount of thermal energy applied. Because of its low vapor flux and recovery rate, the specific energy consumption of MD is extremely high [20]. For a room-temperature fresh feed, the amount of energy required to attain the required temperature is excessive relative to the quantity of water produced.

Unlike other membrane separation processes, the permeation flux of an MD membrane can be affected by two possible mechanisms, namely the vapor transfer resistance and the temperature polarization effect. Through the first mechanism, mass transfer resistance can be minimized by fabricating membranes with large pores and high porosity, an open-cell pore structure, a thin functional layer, and a small tortuosity. Through the second mechanism, the decrease in flux due to temperature polarization can be mitigated by reducing thermal conductivity across the membrane. Because the thermal conductivity of air inside the membrane pores is considerably lower than that in the polymer matrix, the thermal conductivity can be lowered by increasing the membrane porosity [21].

Therefore, it is important to assess the aforementioned physical and physiochemical parameters to create a foundation for future works in this area. Most of the previous studies and reviews of MD have focused on mass and heat transfer models as well as MD preparation. However, little attention has been paid to design processes and optimization, which depend on the operating parameters.

As a result, apart from previous works, a design algorithm that determines the number of modules and in turn helps to calculate the cost of the process is developed along with reviewing the operating ranges of the design parameters. More importantly, because the influence of different heat transfer coefficients on the design process of the MD system has never been studied in previous studies, in this study, the effect of various heat transfer equations is also evaluated. Meanwhile, the influence of design parameters such as membrane thickness and velocity are studied using the data available in the literature. Finally, the total unit cost of water production using MD for low-grade-energy wastewaters is determined.

## 2. Physical and Physicochemical Properties Influencing MD Design

### 2.1. Physical Properties

#### 2.1.1. Membrane Thickness

Membrane thickness is one of the parameters that directly influences transmembrane flux through the mass transfer resistance coefficient. In an early study on vaporization though porous membranes, M. E. Findley discussed the effect of membrane thickness on membrane face temperature differences without considering mass transfer resistance. However, they stated that the impact was only through boiling point elevation, which is proportional to membrane thickness [6]. The membrane thickness has to be greater than a certain minimum value for the membrane face temperature difference to be greater than a certain minimum value. Moreover, they stated that as the membrane thickness increases, its resistance to mass transfer increases. Nevertheless, in this work they did not discuss how thickness is related to both boiling point elevation and mass transfer [6]. However, in a subsequent experimental study, the effect of membrane thickness, measured as membrane weight per unit area, on the overall mass transfer coefficient was illustrated and discussed.

Apart from that, M. E. Findley et al. and G.C. Sarti et al. provided some clarity on the relationship between membrane thickness and the transmembrane flux, by proposing a mass transfer expression on the basis of a fundamental equation of vapor diffusion through stagnant gas, despite the membrane thickness being fixed in the latter experimental study [22,23]. In the equation proposed by Bird et al., the flux length, which is the membrane thickness in this case, is inversely proportional to the mass transfer through the membrane [24]. According to M. E. Findley et al.’s early study on mass and heat transfer in a porous membrane, the membrane thickness has an inverse exponential effect on the overall mass transfer coefficient. Accordingly, as they increased the membrane thickness from 0.0091 to 0.072 (0.5 to 4 g per 0.12 ft^2^), the overall mass transfer coefficient decreased from 5.5 to 1.75 lb/h/ft^2^. In that study, the membrane thickness had a profound effect as the feed temperature decreased. Even so, it was after R.W. Schofield et al. started to study the heat and mass transfer characteristics of MD that the MD membrane thickness was discussed with respect to both heat and mass transfer.

Thereafter, researchers started to consider membrane thickness as one of the most influential parameters in the MD process in terms of its inverse relationship with both mass transfer and heat transfer. In a study on the potential of MD for seawater desalination, S. Al-Obaidani et al. illustrated the effect of the membrane thickness on mass transfer by using four different types of membranes, namely MD020CP2N, MD080CO2N, Home-made, and MD020TP2N, with thicknesses of 0.65, 0.65, 0.05, and 1.55 mm, respectively. It was reported that the transmembrane flux decreased rapidly as the membrane thickness increased and vice versa owing to the inverse proportional relationship [11]. As a matter of fact, the flux decreased by approximately 70% when the membrane thickness increased from 0.25 to 1.55 mm. However, there existed a tradeoff relationship between the transmembrane flux and thermal efficiency with respect to the membrane thickness. The rate of decrease in transmembrane flux and the rate of increase in thermal efficiency was not equal because the flux was directly affected but the thermal efficiency was indirectly affected. The thermal efficiency increased gradually as the membrane thickness increased during operation at temperature gradients exceeding 10 °C. It can be said that the efficiency increased because the decrease in heat conduction through the membrane was always more significant than the decrease in vaporization heat owing to the flux decline at higher membrane thicknesses. To optimize the membrane thickness, M. Khayet recommended the use of a multi-layered membrane endowed with a high level of mass transfer by thinning the hydrophobic layer to the greatest extent possible. Additionally, it was recommended that a low level of heat transfer can be achieved by increasing the overall membrane thickness to the greatest extent possible [5]. In another study, a new type of porous composite hydrophobic/hydrophilic flat sheet membrane for DCMD application was proposed, and the thickness of the hydrophobic layer was determined to be 8 μm, which is an order of magnitude lower than that of the PTFE layer in commercial membranes [25]. The hydrophilic layer is easily wetted by the permeate-side water, which reduces the functional layer thickness and transport distance of water vapors in MD processes. A multilayered or composite membrane for distillation was developed and patented in the early 1980s by D. Cheng. This membrane was composed of a thin lyophobic microporous layer, in which evaporation and condensation occur, and a thin lyophilic layer that prevents the intrusion of the distilland into the pores of the lyophobic layer [9,26,27].

#### 2.1.2. Range of MD Thickness in MD Studies

The range of membrane thicknesses used in various studies differs depending on the type of polymer material, crystallinity, and production method [21]. The range is so wide that it is difficult to identify the optimum thickness that maximizes performance or why a specific membrane thickness is selected in a given study. Nevertheless, in most of the studies, the membrane thickness is between 50 and 180 μm. For example, J. Phattaranawik et al. used 64–77-μm-thick PFTE membranes and 116–126-μm-thick PVDF membranes [28]. By contrast, S. Al-Obaidani et al. used 65–155-μm-thick membranes [11]. P. Termpiyakul et al. used a 126-μm-thick membrane in their heat and mass transfer study [29]. M. Courel et al. used a 30–70-μm-thick PTFE top layer in their 102–165-μm-tick and 110–178-μm-thick TF200 and TF450 experimental composite membranes, respectively [30]. In their review paper, M. Khayet summarized the characteristics of various MD membranes [5]. From the data in their paper, flat sheet membranes made of PTFE/PP composite polymers have a wide range of thicknesses between 8.5 µm and 184 μm [31]. The thickness of the active layer in these composite membranes ranges between 5 μm and 10 μm. For PVDF membranes, the range is relatively narrow between 108 μm and 140 μm. However, in a performance investigation of solar-assisted DCMD, Y.-D. Kim et al. used a 60-µm-thick PVDF hollow fiber membrane [19]. In a composite membrane with a polypropylene (PP) or polyethylene (PE) supportive structure, the thickness ranged from 25 µm to 150 μm. H.C. Duong et al. reported that a 76-μm-thick PP membrane was used in a small-scale seawater desalination plant [32]. However, they also reported a relatively wide range of membrane thickness for capillary and hollow fiber commercial membranes composed of PP as the main material. Accordingly, the membrane thickness range is 25–1550 μm. F. Lagana et al. reported a membrane thickness of 120 μm for a hollow fiber module [18]. For the same module type made of a different material, that is, PE, the thickness range was reported as 55–250 μm. A significantly greater number of PTFE membranes have been developed for hollow fiber and capillary modules relative to flat sheet membranes, and therefore, their thickness ranges from 400 µm to 550 μm.

In the MD literature, R.J. Durham et al. reported the smallest membrane thicknesses of 8.5 and 9.0 μm for Gore-Tex 10387 and Gelman 11104/2TPR membranes, respectively [31].

In sum, the membrane thickness has direct and indirect effects on mass flux. The direct effect stems from its inverse relationship with mass transfer length, whereas the indirect effect stems from its inverse relationship with conductive heat transfer. Therefore, membrane thickness should be analyzed as one of flux-optimizing variables in an MD process. M.E. Findley et al. discussed that heat conduction through a membrane, which is significantly affected by the membrane thickness, should be considered seriously for the following reasons [23]. First, the heat conducted through the membrane is not effective for inducing evaporation, and even the recovery of this heat increases the requirements for heat exchange. Moreover, excessive heat conduction from the membrane interior to the coolant could produce internal condensation, which would probably further increase the membrane heat conductivity and decrease the diffusion or flow of vapors. Eventually, internal condensation might lead to the formation of a continuous liquid channel through the membrane and cause leaking and diffusion of nonvolatile solutes. A third reason why heat conduction should be minimized is that the conducted heat must be transferred through the two liquid films, which increases the temperature drop across these films. Because mass transfer depends on temperatures and vapor pressures at the surface of the membranes, any temperature drop through the liquid films reduces the available driving force for mass transfer.

Hence, there should be a minimum membrane thickness for a given overall or total temperature drop that corresponds to the thickness at which the heat conducted through the membrane is adequate to limit the temperature drop across the membrane to the boiling point elevation.

#### 2.1.3. Pore Size and Porosity

Pore diameter is an important parameter for determining liquid entry pressure (LEP). LEP, also called breakthrough pressure, must always be higher than the applied transmembrane pressure, which is the difference between the applied pressure and membrane face pressure, to prevent liquid solutions from entering the pores. In terms of defining the type of mass transfer in an MD membrane, the pore size is one of the most important parameters in MD processes. MD is only possible with a porous membrane that possesses a nonwettable surface and a sufficiently small pore size such that surface tension forces can withhold liquids from the pores [7,23]. P. Wang et al. reported that to maximize vapor diffusion, permeation flux, and thermal efficiency, the ideal MD membrane should preferably have high porosity and a relatively large pore size in the range of 0.1 μm to 0.3 μm to reduce mass transfer resistance and minimize temperature polarization [21]. Pore size and porosity not only have physicochemical effects on MD processes, but they also affect the physical and structural strength of the membrane. Consequently, a large pore size and high porosity in both the bulk and on the surface degrade the mechanical properties of membranes in both the axial and radial directions. In a study of the transport phenomena in MD, S. Kimura concluded that pore size becomes important when the surface tension of solutions is low [33]. In their experimental study, when a surface-active agent was added to a sodium chloride feed solution to lower the surface tension, both the flux and permeate concentration increased substantially, except at the lowest pore size. Moreover, flux was found to depend on pore size to some extent, whereas permeate concentration did not exhibit any dependency on pore size. Additionally, they demonstrated that the effect of pore size was more significant at lower feed temperatures.

Another factor that affects flux is porosity, which is the number of pores per unit area. Flux has a stronger dependency on porosity than on pore size [33]. S. Kimura et al. reported that as the porosity increased from 0.6 to 0.9, the flux increased from 12 to 18 kg/m^2^/h. S. Al-Obaidani et al. prepared a homemade MD with 45% porosity and compared it to an MD020CP2N membrane with 70% porosity, which is the typical porosity value of commercial MD membranes [11]. According to their study, the commercial MD membrane with higher porosity had a higher flux owing to a higher level of effective molecular diffusion. In addition, it was reported that membranes with a high percentage of porosity or void fraction of the polymer matrix possessed lower thermal conductivity, which increases the thermal efficiency by reducing the loss associated with conductive heat flux. Their calculations indicated that when the membrane porosity was increased by 15%, the vapor flux and thermal efficiency of the MD020CP2N membrane module increased by 26% and 13%, respectively, whereas in the case of the homemade membrane, the corresponding increases were 37% and 3%. The lower values of increase in flux for the second type of membrane may have stemmed from its low thickness.

The pore sizes of the membranes used in an experiment for a modeling study conducted by M. Courel et al. were 0.2 μm and 0.45 μm [30]. F. Lanaga et al. reported a nominal pore diameter of 0.45 μm. R.J. Durham reported a pore size of 0.2 μm [31]. H.C. Duong reported a pore size of 0.3 μm for a PP membrane with 85% porosity. For a shell-and-tube-type hollow fiber membrane module, Y.-D. Kim et al. reported a nominal pore diameter of 0.2 μm and a porosity of 75% [19]. S. Al-Obaidani et al. reported a membrane pore diameter of 0.2 μm for all the four types of PP membranes under investigation [11].

#### 2.1.4. Length of Module and Membrane Area

One of the least-studied optimization parameters that affects the MD process is the module length. It has a positive effect on increasing production capacity but a negative effect on increasing energy loss through heat transfer. In most studies, the module lengths used have been less than or equal to 1 m. For example, the range of membrane lengths used in a thermal sensitivity and cost estimation study by S. Al-Obaidani et al. was 0.24–1 m for different membrane types [11]. For the MD020CP2N and MD080CO2N membranes with lengths of 0.45 and 1 m, respectively, the flux and feed temperature decreased along the axial direction as the permeate temperature increased, while the boiling point elevation increased. However, interestingly, despite the difference in packing density, the difference in flux at the inlet and outlet for the two membranes of different lengths was not significant. Y.-D. Kim et al. discussed effect of the fiber length on the hydrostatic pressure of the bulk feed and permeate. They reported that the feed and permeate sides exhibited hydrostatic pressure drops of 36 kPa and 31 kPa, respectively, for a module length of 0.4 m [19]. For a Microdyn MD020CP2N membrane, F. Lagana et al. reported a module length of 0.45 m [18]. In their MD modeling study, the module length was fixed, and its effect was not discussed further. In a microstructure optimization study of the hollow fiber design, X. Yang et al. performed CFD modeling with an overall length of 0.25 m [34]. In their study, the effects of the module length on both feed- and permeate-side heat transfer coefficients were analyzed. They found that the feed-side heat transfer coefficient distribution curves exhibited a general decreasing trend along the fiber length. For a nonmodified original fiber, as the fiber length increased to 0.25 m from the feed entrance to the feed outlet, the heat transfer decreased from approximately 3500 W/m^2^/K to 1500 W/m^2^/K. However, when the fiber was modified by incorporating various surface microstructures, the rate of reduction decreased owing to the slower buildup of a liquid boundary layer along the feed flow direction, which was attributed to disturbances created by the surface microstructures. Membrane length was found to have a similar effect on the permeate-side heat transfer coefficient, which continued to decrease relative to the inlet-side heat transfer coefficient, except that the rate of decrease was marginally smaller. However, different from the feed-side coefficient, here, too, it appeared to plateau at the outlet.

Few studies have explicitly mentioned the effect of the membrane area, unless it is within the membrane length and the number of fibers or so. According to D. Winter et al., an increase in the membrane surface area does not influence the total distillation output rate. However, it significantly lowers the specific energy consumption [16].

#### 2.1.5. Velocity and Packing Factor

Velocity is the other most important parameter that affects the process performance of MD. In their modeling study of a DCMD system, F. Lagana et al. tried to demonstrate the effect of the transmembrane flux velocity in terms of the feed- and permeate-side flowrates. The rate of change in flux with respect to the feed flowrate was found to be more significant on the permeate side, despite the fact that the flowrate on the permeate side was higher than that on the feed side [18]. Accordingly, when the feed flow rate was decreased from 250 to 100 L/h, the flux decreased from 1.5 to 1.13 kg/m^2^/h; the permeate-side flowrate exhibited a similar change, but the flux decreased from 1.5 to 1.4 kg/m^2^/h.

In their work on the thermal efficiencies of various MD systems, S. Al-Obaidani et al. studied the effects of feed flow velocity on the DCMD flux for an MD020CPN module and a fixed permeate velocity [11]. The simulation and experimental results indicated that both the transmembrane flux and process thermal efficiency increased as the feed flow velocity increased. Quantitatively, the flux increased by 24% and 38% at the feed temperatures of 40 °C and 60 °C when the feed flow velocity increased from 0.2 to 1 m/s. Apparently, the effect was more pronounced for high-temperature feeds. However, when the feed flow velocity exceeded 1 m/s, there were no significant benefits in terms of the transmembrane flux, and they recommended that for MD processes involving hollow fiber modules, the feed flow velocity should be within 0.9–1 m/s. Additionally, they demonstrated that the effect of velocity on the transmembrane flux mainly manifested as an improvement in thermal efficiency, where the membrane face temperature was supposedly close to the feed temperature. In a modeling study of water transportation during asymmetric MD, M. Courel et al. analyzed the effect of the velocity on the membrane flux. Although they found a positive relationship between velocity and the transmembrane flux, the velocity effect plateaued after 1.7 m/s [30]. Their study was conducted using TF200 and TF450 membranes.

P. Termpiyakul et al. studied velocity and reported an increase in the transmembrane flux with velocity [29]. However, the velocity values used in this study (1.85, 2.78, and 3.7 m/s) were higher than those in the previous studies. Similar to the other aforementioned studies, a positive relationship between the velocity and the transmembrane flux was found. Although the increase in flux was significant for the first velocity value, it was insignificant as the velocity was increased to 2.78 and 3.75 m/s. These results illustrated that an increase in velocity increased the heat transfer coefficient through Reynolds number, which resulted in an increase in the membrane surface temperature and membrane polarization coefficient. As the velocity was increased from 1.85 to 2.78 and 3.7 m/s, the heat transfer coefficient increased from 14.018 to 19.417 and 24.407 W/m^2^/K. J. Phattaranawik et al. discussed the effect of velocity on the heat transfer coefficient for both laminar and turbulent flows and reported that the heat transfer coefficient increased with velocity, and the rates of increase were virtually similar for both flow regimes [28].

In a slightly different vein, S. Kimura et al. studied the effect of the feed velocity on the transmembrane flux and found that velocity was inversely proportional to performance ratio, which is permeate flow rate per energy consumption [33]. This indicated that at high velocities, the performance ratio increased owing to better temperature change, which, in turn, reduced the flux. Moreover, they showed that the module length significantly influenced the velocity–flux relationship; as the module length increased, the flux reduced, but the rate of reduction was lower at higher module lengths.

In an experimental study involving a commercial PTFE DCMD process, H. J. Hwang et al. found that high fluxes were exhibited during operations at higher temperatures and velocities [35]. As the velocity was increased from 0.17 to 0.55 m/s, the flux increased from 20 to 30 L/m^2^/h at a feed temperature of 70 °C. Moreover, they demonstrated that the effect of velocity grew stronger as the feed temperature increased. The flux changed little at the feed temperature of 40 °C.

A Boubakri et al. reported the effect of velocity with respect to the Reynolds numbers of 992–16,865. They observed that the permeate-side flux increased with increasing Reynolds number [36]. The increment was parabolic and significant in the initial stages for Reynolds numbers of up to 4000. At Reynolds numbers greater than 4000, the permeate-side flux increased marginally and then reached an asymptotic value. Apparently, increasing the Reynolds number improved the mass transfer coefficient of the interface membrane-solution, in addition to reducing the temperature and concentration polarization effects, which in turn, maintained a large temperature difference.

The effect of velocity on the heat transfer coefficient grew noticeably stronger depending on the employment of spacers, in which the latter shows an improved flux. L. Martinez et al. reported that for a fluid velocity of 0.35 m/s, the heat transfer coefficient values were 3000 and 10,000 W/m^2^/K for the module with open channels and the module including spacers, respectively [37]. At zero concentration, the flux through the open TF200 module was 2.60 × 10^−3^ kg/m^2^/s, whereas for the same module with spacers, the flux increased to 3.8 × 10^−3^ kg/m^2^/s. The difference between the fluxes of the two modules remained similar as the concentration increased.

Inner and outer diameter, packing density, and shell diameter are the other parameters that affect the process design of hollow fiber MD, albeit indirectly through velocity.

X. Yang et al. studied the effect of velocity in terms of the Reynolds number with respect to the temperature polarization coefficient (TPC) and the transmembrane flux [34]. They reported that the average TPC increased with increasing the feed-side Reynolds number, *R_ef_*, for either configuration owing to the improved fluid dynamics at higher flow velocities. Moreover, modified membrane configurations with corrugated surfaces yielded superior results with reduced thickness of the liquid boundary enhanced by increasing velocity. The original module exhibited lower average TPC values, which fluctuated within 6% as the *R_ef_* increased from 420 to 1500; under the turbulence condition with *R_ef_* equal to 2500, the TPC increased by 30%.

### 2.2. Physicochemical Properties

#### 2.2.1. Solute Concentration

In their original study, G.C. Sarti et al. found a slight inverse relationship between flux and solute concentration. However, in their latest low-energy membrane desalination study, G.C. Sarti et al. identified the significant effects of concentration and flux with respect to temperature. The latter study was conducted using pure water to 0.9 M concentrations, and it was reported that the change in flux was affected by the concentration; as the concentration increased, the flux and rate of change in flux increased, especially for concentrations exceeding 0.5 M [7,22].

In their experimental study of the factors that affect the flux of MD on a Durapore 0.45-μm membrane, Scofield et al. stated that flux was affected by the presence of solute [38]. Moreover, they reported a decrease in the activity coefficient with increasing concentration, by showing that the reduction in flux as the concentration increased from 0 to 0.047 M was less than that as the concentration increased from 0.047 to 0.095 M. However, as the feed temperature increased, the difference between the fluxes at higher and lower concentrations increased. This was demonstrated through experiments that were conducted with salt solutions saturated at the feed temperature, resulting in the formation of a layer of precipitated salt crystals on the membrane surface, which reduced the film heat transfer coefficient considerably and led to a rapid decay in flux. At the end of the experiment, it was possibly to remove the salt scales as sheets.

Through concentration, the relative importance of other factors such as viscosity were determined. Roughly one-third of the flux reduction was reportedly caused by viscosity change. An increase in viscosity was found to apparently affect the Reynolds number and reduce turbulence. The vapor pressure at the membrane wall decreased by 25% with respect to that of pure water. Moreover, the heat transfer coefficient decreased by 30%, reflecting a reduced change in vapor pressure. In relation to solute concentration, the only factor that led to an increase in flux was density. An 18% increase in density led to a 14% increase in the film heat transfer coefficient through the Reynolds number, which, in turn, led to a 4% increase in flux. Solute-related factors such as thermal conductivity and heat capacity had a negligible effect on flux, resulting in only a 5% reduction in the heat transfer coefficient. Different from other membrane processes, concentration polarization, too, had a negligible effect on flux.

F. Lagana et al. reported the strong effect of extremely high feed concentrations on the dimensionless resistance of the MD process, which constitutes temperature polarization and concentration polarization and is defined as the ratio of each polarization and membrane resistance [18]. Accordingly, as the concentration increased from 400 g/L to 800 g/L, the dimensionless resistance increased exponentially from 0.8 to 2.25. Moreover, among all the resistances, temperature polarization resistance was affected significantly by concentration changes.

The effect of solute concentration was found to vary with the type of solute. For example, the vapor pressure decreased by less than 3% for a 30% sugar solution, but it decreased by 29% for a 30% salt solution. However, for the same concentration, the effect on viscosity is approximately equal, which indicated that the major cause of flux reduction for sugar solutions was viscosity. However, in a study of the transport resistance in DCMD, L. Martinez et al. reported that a decrease in the flux of a sucrose solution was primarily caused by a decrease in the driving force [13]. Even so, they agreed on the profound effect of viscosity on flux reduction, especially at higher concentrations. A sharp reduction in the feed-side heat transfer coefficient with an increase in sucrose concentration beyond 40 wt% was reported. Accordingly, the predominant flux-controlling resistance was the feed-side film boundary layer resistance. The membranes used in this study were GVHP22 and TF200.

S. Al-Obaidani et al. studied the effect of solution concentration on transmembrane flux for the feed and permeate velocities of 0.39 and 0.28 m/s, respectively. They reported a total flux reduction of up to 50% as the feed concentration increased from 35 to 350 g/L. In addition, thermal efficiency decreased from 58% to 40% [11].

L. Martinez et al. stated that membrane flux was inversely affected by concentration. They described this effect by using different heat transfer and mass transfer resistances [13]. Their results indicated that film resistances, mainly the feed-side film resistance, were the most strongly affected by concentration. Generally, this reduction is a consequence of a decrease in the feed water activity as the amount of feed water increases, which is consistent with the point made by Scofield et al. Moreover, they summarized that if the physical properties of the feed, which affects the film transfer coefficients, do not change considerably as the concentration increases, the resistance does not change significantly with the concentration. Furthermore, the reduction in evaporation efficiency was reported as an effect of a reduction in the feed water activity, which is similar to the results of other studies.

In contrast to the other solutes studied above, M. Courel et al. conducted a study on calcium chloride and reported an almost three-fold increase in transmembrane flux with increasing solute flux. As the solute concentration increased from 30 to 45 wt.%, the solute flux increased from 4 to 11 kg/m^2^/h. There is no explanation as to why the flux increases differently with solute concentration [30].

In a study of the heat and mass transfer characteristics of a DCMD, P. Termpiyakul et al. briefly discussed the effect of feed concentration on the transmembrane flux and reported that it decreased with an increasing feed concentration because of a reduction in vapor pressure. The flux was found to decrease with time as the feed concentration increased, and this effect was significant at high concentrations. At the feed temperature of 40 °C, as the feed concentration increased from 17,500 ppm to 35,000 ppm, the flux decreased from 8.5 to 6.5 kg/m^2^/h [29].

S. Kimura investigated the effect of concentration on the flux for different types of feeds [33]. The results indicated that at high concentration ratios, the fluxes were higher in the MD process than in the RO process. In the process of milk concentration, for a 50% increase, the flux decreased by 150%. Apparently, membranes tend to be fouled because of fat adhesion, which reduces the flux considerably. The concentrations of sugar and gelatin were investigated, and the results indicated a relatively non-steep relationship. In the case of sugar, for a 400% increase in concentration, the flux decreased by only 30%.

D. Winter et al. studied the influence of feed concentration on the total product rate with respect to the mass flowrate and found that the total product rate decreased as the salinity of the feed water increased [16]. The rate of decrease did not differ with the feed flowrate. For example, for the feed flowrate of 200 kg/h, when the feed salinity was increased from 20 to 100 g/kg, the permeate mass rate decreased from 25 to 14 kg/h.

A Boubakri et al. elaborated the inverse relationship between the flux and concentration or ionic strength; as the concentration increased, the dynamic fluid will change as a result of increasing viscosity, and concentration polarization should be added to the temperature polarization, which reduces the imposed DCMD driving force and, consequently, the permeate flux [36]. According to the experimental results of their study, as the ionic strength increased from 0 to 4.3 M, the flux decreased from 3 to 2.4 L/m^2^/h.

H.C. Duong et al. studied the effect of concentration through a different mechanism in the context of recycling brine to achieve high water recoveries [32]. They found that even if recycling increased water recovery, it deteriorated the system performance. Accordingly, they reported that as the water recovery increased from 60% to 70%, the flux decreased from 8 to 7 L/m^2^/h, and there was the risk of scale formation on the membrane surface.

#### 2.2.2. Diffusivity and Mass Transfer Coefficient

In the early 1980s, in studies on thermal membrane processes, mass transport was assumed to be caused by either the Soret effect, according to which flux is induced through thermal diffusion in the bulk liquid phase or through an evaporation-diffusion-condensation process enhanced by capillary forces, or the pervaporation effect, according to which separation is achieved through a partition between a dense membrane and an external vapor phase [7,39]. P. Bellucci et al. studied the mass transport in coarsely porous synthetic membranes and confirmed a close relationship between the flux and temperature gradient [39]. However, according to a thermodialysis study involving PTFE membranes, G.C. Sarti et al. found that the separation in membrane distillation system occurs not because of the Soret effect but rather because of an evaporation-diffusion-condensation process, wherein membrane hydrophobicity plays a crucial role in preventing the liquid phase from entering membrane pores [7]. In a second study involving a water solution as a feed, the process was first investigated experimentally, and it was concluded that a type of low-temperature distillation, enhanced by capillarity, occurred. It was called capillary distillation because the essential prerequisite of the process was the capillary force that prevented the liquid phase from entering the pores.

Under simplified conditions, a liquid–vapor interface is formed on either side of the liquid, which repulses the membrane: typically, evaporation occurs at the interface on the warmer side, and after mass transport through the vapor phase within the pores, condensation occurs at the interface on the colder side. Accordingly, the mass transfer coefficients were expressed based on the theory that water vapor diffuses from the warm side to the cold side through a substantial film of stagnant air [22].

Water flux is expressed under the condition that water vapor diffuses though a substantial film of stagnant air. The diffusion coefficient accounts for both the ordinary and Knudsen transport models [24,40].
(1)$$N=\epsilon M\frac{P_{G}}{RT_{m}}\frac{P_{ht}-P_{cd}}{\tau \delta P_{air\;ln}}$$
where, ϵ is porosity, τ is tortuosity, δ is membrane thickness, $P_{ht}$ and $P_{cd}$ are warm and cold side pressures, $P_{air}$ is the partial pressure of air and $P_{G}$ is the partial pressure of the vapor, $T_{m}$ is the arithmetic mean temperature between the warm side and cold side and M is molecular weight. In the above-referenced studies, even though the diffusion coefficients account for both ordinary and Knudsen transports, it was assumed that the contribution of ordinary diffusion dominated the transport process. The coefficients were primarily estimated based on the gas-phase diffusion coefficients derived from the Stefan–Maxwell hard sphere model and the principle of additive volumes, which was first employed to estimate diffusion coefficients [41]. Then, at low-to-moderate temperature and pressure, they were predicted using the corresponding-states methods. According to Bird et al., in these methods, for binary gas mixtures at low pressures, the diffusion coefficient is inversely proportional to pressure, increases with increasing temperature, and is almost independent of the composition of the given gas pair [24].
(2)$$\frac{pD_{AB}}{{\left(p_{cA}p_{cB}\right)}^{1/3}{\left(T_{cA}T_{cB}\right)}^{5/12}{\left(1/M_{A}+1/M_{B}\right)}^{1/2}}=a{\left(\frac{T}{\sqrt{T_{cA}T_{cB}}}\right)}^{b}$$
which leads to
(3)$$\frac{pD}{T^{b}}=const.$$
where, $D_{AB}$ is binary diffusivity, p is the total pressure, T is the system temperature,$\;p_{cA},\;p_{cB}$ and $T_{cA}$, $T_{cB}$ are critical pressures and temperature of components, respectively, $M_{A}$ and $M_{B}$ are molecular weights of the components, and a and b are dimensionless constants. The above relationship helps correlate the average values of pressure, temperature, and diffusion coefficient with that at any point.

M.E. Findley et al. are among the early researchers in the United States (1969) to study the mass and heat transfer relationships of porous membranes. Their experimental results indicated that the major factor influencing the rates of transfer was diffusion through a stagnant gas in the membrane pores [23]. G.C. Sarti et al. described the mechanism of mass transfer for multicomponent fluids in MD [7,22]. A porous partition in contact with two liquid phases that are stalled at the pore entrances by capillary forces was considered. In this study, the pressure on either liquid was smaller than the capillary pressure of the membrane, that is, the liquid phases were kept out of the pores by the capillary forces. Thus, a gaseous phase was immobilized within the membrane pores, where because of a difference in vapor pressure at the opposite membrane surfaces, evaporation at one end and condensation at the opposite capillary end occurred.

However, M.E. Findley et al. illustrated that the above equation was oversimplified and should be corrected for heat transfer effects. In addition to diffusion, factors such as heat transfer coefficients, membrane thermal conductivity, and temperature difference should be considered when determining the mass transfer coefficient in MD. Accordingly, the following mass transfer coefficient formula was derived:

In general, the mass flux across an MD membrane can be expressed as follows:(4)$$N=C_{m}\left(P_{ht}-P_{cd}\right)$$
where,
(5)$$\frac{1}{C_{m}}=7.406\frac{\;X\;P_{Bavg.}}{{\left(\bar{T}\right)}^{3/4}}+0.00642\frac{\partial P}{\partial T}\Delta H_{v}+3.76\frac{\;P_{Bavg.}}{{\left(\bar{T}\right)}^{3/4}}-0.643\;+0.043\left(\Delta T-E\right)$$
where $C_{m},$ $\bar{T},\;P_{Bavg.}$, E, and X are the membrane mass transfer coefficient, the average temperature on the solution side and condensate side, total pressure minus average of solution side and condensate side vapor pressures, boiling point elevation of solution, and thickness of membrane expressed as weight per area, respectively. The above coefficient was computed by conducting three groups of experiments for determining the effect of vapor pressure difference on evaporation rates, the effect of membrane thickness on mass transfer, and the effect of temperature on air gas partial pressure. The first experiment and determination of mass transfer coefficient ended up short of missing the heat transfer effects on the coefficient, and therefore, it was corrected in the second experiment. Even so, to establish the independent effect of temperature stemming from an external heat source, the third experiment was set up to correlate the equations based on a diffusive mass transfer mechanism though stagnant air.

#### 2.2.3. Mass Transfer Coefficient Models

In recent studies, the mass transfer in MD has been described using the dusty-gas model (DGM), which is the most general model for a flux through a porous medium, where the medium is dust particles. This model was first described by Maxwell for dilute gases. Even though it was derived for isothermal fluxes, it has been successfully applied to non-isothermal systems through the inclusion of thermal diffusion and thermal transpiration terms, which are negligible in the MD operating regime [42]. This model represents a more theoretically sound approach to address diffusion through a porous medium. In its most general form, the DGM applicable to MD (neglecting surface tension) can be expressed as follows:(6)$$\frac{N_{i}^{D}}{D_{ie}^{k}}+\overset{n}{\underset{j=1\neq i}{\sum }}\frac{p_{j}N_{i}^{D}-p_{i}N_{j}^{D}}{D_{ije}^{0}}=\frac{-1}{RT}\nabla p_{i}$$
(7)$$N_{i}^{V}=\frac{-p_{i}B_{0}}{RT\mu }\nabla P$$
where $N_{i}^{D}$ is the diffusive flux, $N_{i}^{V}$ is the viscous flux, P is the total pressure, $p_{i}$ is the partial pressure of component i, and μ is the fluid viscosity; the effective ordinary and Knudsen diffusivities are defined as $D_{ije}^{0}=K_{1}PD_{ij}$ and $D_{ie}^{k}=K_{0}{\left(\frac{8RT}{\pi M_{i}}\right)}^{1/2}$, respectively.

The constants, which were estimated from the membrane pore radius r, tortuosity τ, and porosity ϵ (assuming the membrane consisted of uniform cylindrical pores), $K_{0}=\frac{2\epsilon r}{3\tau }$*,*
$K_{1}=\frac{\epsilon }{\tau }$*,* and $B_{0}=\frac{\epsilon r^{2}}{8\tau }$ were found to depend on membrane geometry and membrane–molecule interactions. These constants are best determined experimentally because the complex geometries of most membranes make it impossible to perform direct calculations.

However, when the more general DGM is broken down, the mass transfer resistance across an MD membrane in a typical MD process can be conveniently described in terms of the serial electrical resistances upon the transfer between the bulks of two phases in contact with the membrane, as illustrated in Figure 1. The mass transfer boundary layers adjoining the membrane generally make negligible contributions to the overall mass transfer resistance. The mass transfer through the membrane can be divided into three models based on collisions between molecules, and/or molecules within the membrane. Molecular diffusion across the polymeric membrane often represents the controlling step. Resistance to mass transfer on the distillate side is omitted whenever MD is operated with pure water as the condensing fluid in direct contact with the membrane, or when the configuration used to establish the required driving force is based on a vacuum [10,30,42].

When the pore size is too small, such that the collision between the molecules and the inside walls of the membrane suitably express the mass transport, that is, $K_{n}>1\;or\;d_{p}<\lambda$ (Knudsen region), the molecular collisions can be ignored, and the resistance can be expressed as follows:(8)$$C_{Kn}=\frac{2\pi }{3}\frac{1}{RT}{\left(\frac{8RT}{\pi M_{w}}\right)}^{0.5}\frac{r^{3}}{\tau \delta }$$

When the molecules travel corresponding to each other under the influence of a concentration gradient, that is, $K_{n}\langle 0.01\;or\;d_{p}\rangle 100\lambda$ (continuum region), the mass transfer resistance can be expressed as follows:(9)$$C_{D}=\frac{\pi }{RT}\frac{PD}{P_{air}}\frac{r^{2}}{\tau \delta }$$

Among the specific models, K. W. Lawson et al. and other reviewers of the process suggested that for de-aerated DCMD, the reduced Knudsen–molecular diffusion transition form of the DGM should be applied to describe the vapor flux of pure water across a membrane [10,11,42]. It is expressed as follows for $0.01<K_{n}<1\;or\;\lambda <d_{p}<100\lambda$ (transition region):(10)$$C_{c}=\frac{\pi }{RT}\frac{1}{\tau \delta }\;{\left[{\left(\frac{2}{3}{\left(\frac{8RT}{\pi M_{w}}\right)}^{0.5}r^{3}\right)}^{-1}+{\left(\frac{PD}{P_{air}}r^{2}\right)}^{-1}\right]}^{-1}$$
where $D=4.46\times 10^{-6}\frac{\varepsilon }{\tau }T_{avg}^{2.334}$; $P_{air}$ is the partial pressure of the air in the pores, P is the total pressure inside the pore, λ is the mean molecular free path, $K_{n}$ is the Knudsen number. The above equations can be derived from the equation of the DGM model.

#### 2.2.4. Experimental Values of Mass Transfer Coefficients

P. Termpiyakul et al. experimentally verified the mass transfer coefficient from a flux versus vapor pressure difference graph and computed the mass transfer coefficient, which was found to be dependent on membrane characteristics and vapor properties, such as porosity and tortuosity; the computed value was 0.0024 kg/m^2^/h/Pa. In a study of heat and mass transfer in MD, R.W. Schofield computed the mass transfer coefficient as 0.00306 kg/m^2^/h/Pa or 8.5 × 10^−7^ kg/m^2^/s/Pa on the basis of Hanbury and Hodgkiess analysis [40,43]. It was determined from the slope of a plot of temperature over flux vs. change in pressure over change in temperature.

In a study investigating the effect of the MD mass transfer coefficient C on the performance enhancement of non-baffled and baffled modules, H. Yu et al. found that the TPC decreased significantly with an increasing C value, regardless of the existence of baffles, signifying a loss of the overall driving force [44]. However, a higher C value compensated for this loss, and the mass flux showed an increasing trend. A membrane with a lower C value was found to be less vulnerable to the TP effect. In this case, the introduction of turbulence aids such, as baffles did not have a substantial effect in terms of improving the system performance. By contrast, the introduction of baffles into the module greatly enhanced the mass flux and TPC for a membrane with a high C value, where the main heat transfer resistance was determined by the fluid-side boundary layers. In this study, the mass transfer coefficient values ranged from 2.0 × 10^−7^ to 1.0 × 10^−6^ kg/m^2^/s/Pa.

H.J. Hwang et al. investigated the effect of velocity on the mass transfer coefficient under different temperature conditions and reported an increase in the mass transfer coefficient with velocity, because of a decrease in temperature polarization owing to increases in the flow rate and Reynolds number, and a decrease in the boundary layer thickness. This influence was significant as the feed temperature increased. The experimental results indicated that at the feed temperature of 60 °C, as the velocity increased from 0.15 to 0.55 m/s, the mass transfer coefficient increased from 0.003 to 0.0042 L/m^2^/h/Pa. By contrast, at the feed temperature of 40 °C, it only increased from 0.0028 to 0.003 L/m^2^/h/Pa. In this study, the effect of velocity on flux was studied independently; it was found that likewise, the flux improved as the velocity increased, especially at higher feed temperatures. At low feed temperatures, the effect of velocity on mass transfer was less significant than that on the overall flux. Another important consideration in this study was the effect of velocity on the flux with respect to flow arrangements, such that for a 0.15 m × 0.4 m module, the flow was arranged along both the width and length. As a result, the flow along the width, that is 0.15 m, had a higher permeate flux under the same velocity conditions because the length or water path was shorter and, therefore, there was a lower temperature drop along the membrane in this configuration. Moreover, velocity was found to have an indirect effect on pressure and must not exceed the tested membrane LEP value to prevent permeation of the salt solution.

#### 2.2.5. Air Partial Pressure

Based on the results of an experiment, R.W. Schofield et al. reported that fluxes increased by up to 50% owing to deaeration, and the membrane permeability and feed-side heat transfer coefficient increased by around seven times. This reduced conduction heat loss amounted to less than 10%. However, the downside was that the deaeration caused temperature polarization, which reduced the thermal driving force by five times. In this study, the influence of air partial pressure inside the membrane pores on the flux was reported with respect to the heat transfer coefficient, and this influence was significant at higher heat transfer coefficients. For example, when the air partial pressure increased from 20 to 100 kPa, the flux decreased from 30 to 25 kg/m^2^/h for a heat transfer coefficient of 500 W/m^2^/K and from 160 to 80 kg/m^2^/h for 5000 W/m^2^/K.

#### 2.2.6. Thermal Conductivity

Thermal conductivity is an important characteristic property in MD owing to its direct influence on heat loss through the membrane material from the hot side to the cold side. Studies have focused on compositing less thermally conductive polymers with hydrophobic ones to reduce heat loss and improve process efficiency. P. Wang et al. tabulated the characteristic properties of commercial polymer materials commonly used to prepare MD membranes [21]. PTFE, PP, PE, PVDF, and Hyflon polymers possess thermal conductivities of 0.25, 0.17, 0.40, 0.19, and 0.2 W/m/K, respectively. Perhaps thermal conductivity is one of the reasons why PP is composited with PTFE. Y.-D. Kim et al. reported a thermal conductivity of 0.25 W/m/K for PTFE membrane material [19].

#### 2.2.7. Heat Transfer Coefficient

The main concern of early MD developers was whether the system is economically competitive, followed by its technical feasibility. Therefore, the first step was to measure the heat transfer coefficients, where most of the values were obtained from laboratory experiments, and energy, membrane, and plant volume cost were interrelated. In those studies, heat transfer coefficients were expressed based on the feed temperature, rather than the velocity or channel geometry.

In 1985, an MD assessment was conducted by W.T. Hanbury et al. at a constant permeate temperature by increasing the feed temperature. The following empirical relationship between the heat transfer coefficient and the average temperature between the feed and permeate was formulated from the experimental results [43]:(11)$$h\left(T\right)=0.0049T^{1.17}\;\left[\mathrm{kW}/K/m^{2}\right]\;\;T\;\mathrm{is}\mathrm{in}\;{}^{\circ}C$$

According to the aforementioned study, the heat transfer coefficient in MD was substantially lower than the equivalent values normally achieved in distillation plants, which indicated that the surface requirements for the process were likely substantial.

Moreover, it was suggested from the outset that for the MD process to become competitive for seawater desalination, one or both of the following conditions must be fulfilled:The price of the membrane must decrease by at least an order of magnitude;Considerably higher distillation heat transfer coefficients must be realized.


In the initial membrane distillation studies, the heat transfer coefficients and Nusselt number were expressed based on the Raleigh number Ra, which is a product of the Grashof and Prandtl numbers, as follows:(12)$$Nu=1+1.44\left(1-\frac{1708}{Ra}\right)+\left[{\left(\frac{Ra}{5830}\right)}^{1/3}-1\right]$$

A variety of empirical correlations have been derived from the literature to evaluate the boundary layer heat transfer coefficient. The following correlations were selected from the proposed ones, as suggested in previous studies [10,12,28,29]:

For laminar flows:(13)$$Nu=0.074Re^{0.2}{\left(GrPr\right)}^{0.1}Pr^{0.2}$$
(14)$$Nu=0.0974Re^{0.73}Pr^{0.13}$$
(15)$$Nu=0.13Re^{0.64}Pr^{0.38}$$
(16)$$Nu=1.62Re^{0.2}{\left(RePr\left(d_{h}/L\right)\right)}^{0.33}$$
(17)$$Nu=0.664Re^{0.5}Pr^{0.33}{\left(2\left(d_{h}/L\right)\right)}^{0.5}$$

For turbulent flows:(18)Nu=0.023(1+6(dhL))Re0.8Pr1/3
(19)$$Nu=0.036Re^{0.8}Pr^{1/3}{\left(d_{h}/L\right)}^{0.055}$$
where Re, Pr, $d_{h}$, L are the Reynolds number, Prandtl number, hydraulic diameter, and module length, respectively.

## 3. Methods

This section may be divided by subheadings. It should provide a concise and precise description of the experimental results and their interpretation, as well as the experimental conclusions that can be drawn.

### 3.1. Mass Flux and Threshold Temperature

In MD, the mass flux is assumed to be proportional to the vapor pressure difference across the membrane, which is a function of the temperatures on the feed and permeate sides [7,12,40]. However, on each side, the real temperature gradient, which is the membrane face temperature gradient, is different from the bulk temperature gradient. The former is lower because of temperature polarization, as illustrated in Figure 2.

The mass flux is expressed in terms of the feed- and permeate-side vapor pressures $P_{2}$ and $P_{3}$, respectively, and the mass transfer coefficient $C_{md}$, as follows:(20)$$J=C_{m}\left(P_{2}-P_{3}\right)$$

For pure water or a very diluted solution, when the temperature difference across the membrane surfaces is less than or equal to 10 °C, the mass flux can be expressed using the following simplified formula:(21)$$J=C_{m}\frac{dP}{dT}\left(T_{f,m}-T_{p,m}\right)$$
where $\frac{dP}{dT}$ is calculated from the Clausius–Clapeyron equation at the average membrane temperature. $T_{f,m}$ and $T_{p,m}$ are feed-side and permeate-side membrane temperatures, respectively. However, for a concentrated or multicomponent solution, the reduction in vapor pressure due to dissolved species must be considered (derivation in Appendix A).
(22)$$\;J=C_{m}\frac{dP}{dT}\left[\left(T_{f,m}-T_{p,m}\right)-\Delta T_{th}\right]\left(1-x_{m}\right)$$
where the threshold temperature is $\Delta T_{th}=\frac{RT^{2}}{M_{w}\Delta H_{v}}\frac{x_{f,m}-x_{p,m}}{1-x_{m}}$ and $x_{f,m}$ and $x_{p,m}$ are the feed-side and permeate-side membrane concentration, $x_{m}$ mean concentration, and $\Delta H_{v}$ are the heat of vaporization.

For simplicity, the following expression can be used to express the vapor pressures: [32].
(23)$$P^{*}=exp\left(23.1964-\frac{3816.44}{T-46.13}\right)$$

For a saline solution, the presence of salts in the solution reduces the water activity level, and therefore, the water vapor pressure. Thus, the partial vapor pressure of water on the membrane surfaces in the DCMD of saline solutions (*P*) is calculated as follows:(24)$$P=x_{water}\left(1-0.5x_{water}-10x_{salt}^{2}\right)P^{*}$$
where $x_{water}$ and $x_{salt}$ are the mole fractions of water and salt, respectively.

### 3.2. Heat Transfer

As shown in Figure 2, the heat transfer from the feed side to the permeate side is determined based on the temperature gradient across the thin-film boundary layers on both faces of the membrane and the membrane itself. As expressed in Section 3.1, the actual vapor pressure gradient that affects the mass flux must be calculated based on the temperature values at the membrane wall, which in turn, must be determined from feed and permeate bulk temperatures, latent heat, and conductive and convective heat transfer equations on the film boundary layer. In MD, the heat flux must be reduced to exploit the enthalpy of vaporization. Therefore, to maintain the membrane face temperatures close to the bulk temperatures, the heat transfer coefficients should be adequately high.

Latent heat transfer from vapor flux is expressed as follows:(25)$$Q_{v}=J\Delta H_{v}=C_{m}\frac{dP}{dT}\Delta H_{v}\left(T_{f,m}-T_{p,m}\right)$$

Conductive heat transfer across the membrane is expressed as follows:(26)$$Q_{c}=\left(k_{m}/\delta \right)\left(T_{f,m}-T_{p,m}\right)$$
where the effective thermal conductivity of the membrane is calculated from the gas and solid conductivities as $k_{m}=\epsilon k_{g}+\left(1-\epsilon \right)k_{s}$. Here, $k_{g}$ and $k_{s}$ are the vapor and membrane conductive coefficients, respectively.

Therefore, the total heat transfer flux inside the membrane is given as follows:(27)$$Q_{m}=\left(k_{m}/\delta \right)\left(T_{f,m}-T_{p,m}\right)+J\Delta H_{v}$$

The heat transfer equations outside the membrane on the feed and permeate boundary layers are expressed as follows:(28)$$Q_{f}=h_{f}\left(T_{f}-T_{f,m}\right)$$
(29)$$Q_{p}=h_{p}\left(T_{p,m}-T_{p}\right)$$
where, $h_{f}$ and $h_{p}$ are feed-side and permeate-side bulk heat transfer coefficients, and $T_{f}$ and $T_{p}$ are feed-side and permeate-side bulk temperatures, respectively.

In the steady state, all heat transfer fluxes through the boundary and membrane walls, along with the overall heat transfer flux, contribute equally to deriving and determining the membrane wall temperatures, as follows:

From the feed-side film and membrane wall equations, we have:(30)$$h_{f}\left(T_{f}-T_{f,m}\right)=\alpha \left(T_{f,m}-T_{p,m}\right)+J\Delta H_{v}$$
where $\alpha =\left(k_{m}/\delta \right)$

From the feed- and permeate-side film equations, we have:(31)$$h_{f}\left(T_{f}-T_{f,m}\right)=h_{p}\left(T_{p,m}-T_{p}\right)$$

From this equation, $T_{p,m}$ is expressed as follows:(32)$$T_{p,m}=\beta \left(T_{f}-T_{f,m}\right)+T_{p}$$
where β=hphf

Then, by substituting Equation (32) into Equation (31), and rearranging, we have:(33)$$h_{f}T_{f,m}=h_{f}T_{f}-J\Delta H_{v}-\alpha \left(T_{f,m}-\beta \left(T_{f}-T_{f,m}\right)-T_{p}\right)$$
(34)$$T_{f,m}=\frac{\alpha \left(T_{p}+\beta T_{f}\right)+h_{f}T_{f}-J\Delta H_{v}}{h_{f}+\alpha +\alpha \beta }$$

Similarly,
(35)$$T_{p,m}=\frac{\alpha \left(T_{f}+\beta^{-1}T_{p}\right)+h_{p}T_{p}+J\Delta H_{v}}{h_{p}+\alpha +\alpha \beta^{-1}}$$

### 3.3. Design of MD Module

#### 3.3.1. Material Balance for Counter-Flow Hollow Fiber Module (Feed Inside Fiber)

The differential material balance in hollow fiber modules is expressed based on the packing factor θ, which is the fraction of the area of the fiber bundle to the shell inside the area of the module. Another important variable is ζ, which is the surface area to volume ratio of the module. There exist two types of models for calculating the material balance cross flow and parallel flow.

In the cross-current arrangement as illustrated in Figure 3, the feed flows radially over the outside surface of the fiber bundle, and the condensed permeate flows inside the fiber laterally along with the coolant. The material balance for the cross-flow arrangement is expressed as follows:

Permeate-side balance (inside a fiber):$${2\pi r\Delta r\theta v_{p}|}_{z+\Delta z}-{2\pi r\Delta r\theta v_{p}|}_{z}=2\pi r\Delta r\Delta z\zeta \left(J/\rho \right)$$
(36)$$\frac{dv_{p}}{dz}=\left(J/\rho \right)\frac{\zeta }{\theta }$$

Feed-side balance:$${\rho_{f}2\pi rLv_{f}|}_{r+\Delta r}-{\rho_{f}2\pi rLv_{f}|}_{r}=-\rho_{p}2\pi r\Delta r\theta {v_{z}|}_{z=L}$$
$$\frac{d\left(rv_{f}\right)}{dr}=-\frac{\theta r}{\rho_{f}L}{v_{z}|}_{z=L}$$
(37)$$\frac{dv_{f}}{dr}=-\frac{v_{f}}{r}-\frac{\theta r}{\rho_{f}L}{\rho_{p}v_{z}|}_{z=L}$$

Feed-side component balance:$$\rho_{f}\frac{d\left(rv_{f}c\right)}{dr}=0$$
$$c\frac{d\left(rv_{f}\right)}{dr}+rv_{f}\frac{d\left(c\right)}{dr}=0$$
(38)$$\frac{d\left(c\right)}{dr}=-\frac{c\theta }{v_{f}L}{v_{p}|}_{z=L}$$

In the counter-current arrangement as illustrated in Figure 4, the feed stream is inside the fiber, and the permeate stream is washed away by the coolant fluid laterally in the opposite direction. The material balance for the counter-current flow arrangement is expressed as follows:

Feed-side balance:$${2\pi r\Delta r\theta v_{f}|}_{z+\Delta z}-{2\pi r\Delta r\theta v_{f}|}_{z}=-2\pi r\Delta r\Delta z\zeta \left(J/\rho_{f}\right)$$
(39)$$\frac{dv_{f}}{dz}=-\left(J/\rho_{f}\right)\frac{\zeta }{\theta }$$

Feed-side component balance:$$\rho_{f}\frac{d\left(v_{f}c\right)}{dz}=0$$
$$c\frac{d\left(v_{f}\right)}{dz}+v_{f}\frac{d\left(c\right)}{dz}=0$$
(40)$$\frac{d\left(c\right)}{dz}=\frac{c}{v_{f}}\left(J/\rho_{f}\right)\frac{\zeta }{\theta }$$

Permeate-side balance:$${2\pi r\Delta r\left(1-\theta \right)v_{p}|}_{z}-{2\pi r\Delta r\left(1-\theta \right)v_{p}|}_{z+\Delta z}=2\pi r\Delta r\Delta z\zeta \left(J/\rho \right)$$
(41)$$\frac{dv_{p}}{dz}=\left(J/\rho \right)\frac{\zeta }{\left(1-\theta \right)}$$
where θ is the packing factor and ζ is the surface area to volume ratio.

#### 3.3.2. Design Algorithm

To our knowledge thus far, a design algorithm for an optimized MD process has not been proposed in the literature. Therefore, we propose an algorithm based on an adjustment of the number of modules in parallel, with respect to the target production capacity and the calculated production as shown in Figure 5. The design process is executed based on the membrane types, membrane properties, and the design parameters given in a previous study [11]. It was assumed that the distribution of the fiber diameter in MD modules is uniform. Given the membrane and feed parameters, the first step in the algorithm is to calculate the membrane face temperature. For simplicity, the number of modules in the series, feed velocity, and permeate velocity are fixed as 1, 0.2 m/s, and 0.28 m/s, respectively, and the change in temperature along the stream is assumed to be negligible. The membrane wall temperature is evaluated iteratively because initially, some of the parameters such as enthalpy, are calculated based on the bulk temperature. Thereafter, the production capacity from the permeate stream is calculated and compared to the target production capacity, and the number of modules is adjusted based on the sign of the difference between the target and calculated capacity values. This is a simplified algorithm with only one modification in the number of modules in parallel. However, any change in the number of modules in the series, number of fibers, and module length, which affects the change in temperature along the length, must be included in future studies. Then, a cost analysis is performed based on the total membrane area. Finally, the energy consumption is calculated based on the feed temperature and the feed mass flowrate and is then simply integrated with the temperature function of the specific heat capacity.

### 3.4. Mass Flux and Threshold Temperature

Apart from other membrane processes, in MD, parameters such as density, specific heat capacity, viscosity, and thermal conductivity are affected by both the temperature and concentration. In a study on the thermophysical properties of seawater, M.H. Sharqawy et al. reviewed and collated correlations and dates for the aforementioned properties and more, on the basis of independent variables such as concentration and temperature [45]. The correlations were obtained from available tabulated data of the properties. For solutions other than seawater, the parameters were computed and compiled using a collection of general physical property estimation formulas reported in various studies [10,40,46,47,48,49]. The constants in the correlations vary based on the type of solute used. The two most important parameters are vapor pressure and enthalpy of vaporization, which are estimated as follows:(42)$$P_{v}=\exp\left(23.238-\frac{3841}{T_{m}-45}\right)$$
(43)$$\Delta H_{v}=1.7535T+2024.3$$

#### Heat Transfer Coefficients

It is important to estimate heat transfer coefficients in the MD process because of the profound effect of thermal boundary layer resistances on the differences between the bulk temperature and the membrane interface temperature. The feed-side membrane solution interface temperature is lower than the bulk temperature, whereas the permeate-side temperature is higher, which is analogous to the ICP and ECP effects in the FO process. This effect reduces the vapor pressure difference, which in turn, reduces the mass flux and the efficiency of the system. This phenomenon is expressed in terms of TPC, which is often used as an indirect index for the efficiency of the MD process, and its value ranges between 0.4 and 0.7.
(44)$$\mathrm{TPC}=\frac{T_{f,m}-T_{p,m}}{T_{f}-T_{p}}$$

### 3.5. Estimating Design Parameters

The design and cost analysis of MD systems is mainly based on the membrane area for two reasons. First, because the feed is supposed to be from domestic or factory wastes at a moderately elevated temperature, the cost of energy is omitted. Second, the mass flux of MD systems is lower, especially in this particular study, where the feed temperature is 40 °C. Therefore, to achieve the target production capacity, the required membrane area is large.

The design parameters and cost calculations are based on previous studies as shown party on Table 1 [11,30,50,51,52]. The design parameters tabulated herein are exclusively those of the homemade module. The membrane cost is updated based on the 2019 Chemical Engineering Index. The waste disposal cost can be omitted because there would be an existing system. The design and cost parameters were summarized in Table 2 and Table 3.

## 4. Results and Discussion

### 4.1. Effect of Membrane Properties

#### 4.1.1. Effect of Membrane Thickness

The results of the feed and permeate velocity $v_{f}$ and $v_{p}$; Reynolds number $Re_{f}$ and $Re_{p}$; interface temperatures and their difference ΔT; and the mass transfer coefficient are presented in Table 4. The process parameter analysis is based on the mass flowrate of the feed $m_{f}$, which is equal to 0.055 kg/s, and the mass flowrate of the permeate $m_{p}$, which is equal to 0.027 kg/s. Equation (13) is used to calculate the heat transfer coefficient for a laminar flow.

In any membrane process, the main result that depicts the process efficiency is the transmembrane flux with respect to the driving force. For the proposed module types, the transmembrane flux increases with the feed temperature as shown in Figure 6. The flux difference between the modules stems from the difference in the thickness of their membranes, which directly affects the coefficient of mass flux $C_{m}$, as expressed in Equation (10). In this study, similar porosity and tortuosity is used for all the membranes to compare the effect of thickness on the membranes. The change in the shell diameter of each membrane type is calculated based on a difference in the outer diameter of the fiber. Accordingly, the homemade module, with the thinnest membrane, provides the best flux for the entire range of feed temperatures. The effect of membrane thickness is given by $C_{m}$ in Table 4, where the transmembrane flux of the homemade modules increases significantly, even though this module has the smallest surface area and the lowest temperature gradient. It has been specified in Table 1 that the membrane thickness for the homemade module is one order of magnitude less than the next lowest membrane thickness. Separately, the results of the other modules clearly highlight the overwhelming influence of membrane thickness on the transmembrane flux relative to those of the other parameters. Thus, even though the difference in membrane interface temperature for MD020TP2N is greater than those for the other modules, its transmembrane flux is the lowest because of its largest membrane thickness, which reduces $C_{m}$. For all the modules, the TPC values are in the optimum range.

The effect of membrane thickness is further illustrated in this study by varying the parameter values. For instance, consider a comparison of the MD020CP2N and MD080CO2N modules by changing the thickness of the first module to 0.05 mm while keeping the fiber outer diameter, packing factor, and membrane length unchanged. The membrane thickness of the second module is kept unchanged, that is, 0.65 mm. Apparently, to maintain the same packing factor for both modules, the fiber outer diameters should be the same, and the change should be made to the fiber inner diameter. The number of fibers should be the same as the given values.

Therefore, for a similar shell inside diameter, with the number of fibers equal to 40, and a packing factor of 0.7, the transmembrane fluxes remain different, as shown in Figure 7. The difference between the transmembrane fluxes decreases because the number of fibers in the second module is reduced from 467 to 40, which brings us to the effect of another parameter, namely velocity. Moreover, the influence of the feed velocity caused by the fiber internal diameter on the transmembrane flux is depicted in Table 4, where despite a lower temperature difference result caused by a lower permeate velocity, the membrane feed temperature of the homemade membrane type is the highest. That is because the feed velocity influences the heat transfer coefficient thought the Reynolds number, which in turn, has an effect on the membrane face temperature.

#### 4.1.2. Effect of Velocity and Number of Fibers

As indicated by the velocity results in Table 5, the feed-side velocity, which flows inside the fiber, is indirectly influenced by the membrane thickness. The velocity value of the first module is lower than that of the second module, even though both modules have the same number of fibers. This is because for the fixed outside fiber diameter, which ensures that the packing density remains unchanged, the membrane thickness inside the fiber changes, which alters the inside fiber diameter. This, in turn, reduces the feed velocity inside the fiber because the thickness decreases to 0.05 mm in case of the first module. The diameter reduction is lower with respect to the second module, which has a membrane thickness of 0.65 mm. Because of the fixed outside diameter, the permeate-side velocity remains unchanged.

In the previous part, the effect of the transmembrane flux is related to different membrane thicknesses for the same number of fibers, that is, 40. However, if the number of fibers in the second module is changed back to 467 and its thickness is changed to 0.05 mm, similar to the case of the first module, the difference between the fluxes of the two modules increases, as depicted in Figure 8, because of a significant reduction in the feed-side velocity or Reynolds number of the second module, which has an effect on the heat transfer coefficient, as listed in Table 6.

### 4.2. Effect of Heat Transfer Coefficient Estimation

Even though previous studies have listed multiple Nusselt number correlations for estimating heat transfer coefficients, it is difficult to find the best correlation for theoretical design, especially in the case of laminar flow, which is considered the typical flow regime in MD. The most commonly suggested correlations are collected and listed for laminar flow (Equations (13)–(17)) and turbulent flow (Equations (18) and (19)). The transmembrane flux results are studied using these correlations for feed velocities of 0.2–1 m/s and the feed temperature of 40 °C [11,53]. In the turbulent region, Equation (18) is used because it is widely recommended.

Figure 9 shows the existence of a significant discrepancy in the flux values with respect to the heat transfer coefficients in both the laminar and turbulent regions. In some of the recommended correlations, the heat transfer coefficients are unaffected by the change in velocity. Therefore, as a recommendation, the heat transfer correlations are one of the most crucial expressions that must be validated experimentally before the design process.

### 4.3. Design and Cost Analysis

In this study, an attempt is made to explore the economic viability of a membrane desalination system for feeds with low-grade heat, such as domestic wastewater from a bathroom, that from a barley sprout cooling spray water, and that from the bottling section of breweries, which are considered to be at close to 40 °C [54,55,56].

The specific energy consumption of the MD system is calculated from the sensible enthalpy required to increase the temperature of fresh feed to the required vapor pressure–temperature.
(45)$$SEC=\frac{m_{f}C_{p}\Delta T}{m_{p}}$$

The total unit product cost results in Table 7 conform to previous recommendations that MD with the available waste heat should be feasible. In this study, the costs of the waste feeds from a domestic bathroom and a brewery are calculated for different flow velocities. It is demonstrated that the total unit product cost is influenced by the feed velocity. Even if the cost values are marginally higher owing to low flux or production rates because of the low-grade heat of the feed, they are 5–20 times lower than the cost of fresh seawater feed at room temperature. These results indicate that the cost of elevating the temperature of fresh feed from 25 °C to 40 °C is higher than the other constituent costs by a wide margin.

## 5. Conclusions

MD is an emerging process for desalination that couples thermal and mass transfer though a membrane. The driving force for transmembrane flux is the difference in partial pressure between the feed and permeate streams, which is a function of the temperature of the two streams.

Most studies on MD have focused on membrane properties, membrane arrangements, mass and heat transfer mechanisms, and thermal efficiency. Even if some of these studies have investigated the influence of operating conditions at the lab scale or in pilot plant setups, little effort has been dedicated to create an algorithmic flow for designing and optimizing the MD process.

Therefore, in this study, a design algorithm was developed and a cost analysis of low-grade-heat domestic feeds, such as brewery factory waste feed, was conducted.

Along with the design process and the design algorithm, the influences of design parameters such as the membrane thickness and velocity were further studied on the basis of the data available in the literature. For a production capacity of 100 m^3^/h, the total unit product costs of USD 1.5897/m^3^, 2.69/m^3^, and 15.36/m^3^ are obtained for low-grade waste desalination, whereas for a fresh cold seawater waste feed, the costs increase significantly to USD 32.84/m^3^, 52.20/m^3^, and 83.32/m^3^ for feed velocities of 0.25, 1, and 2.5 m/s, respectively. The effect of the membrane thickness on the transmembrane is depicted on the mass transfer coefficient, and the homemade membrane type, with the thinnest value, gives the largest flux.

Moreover, the effect of the different heat transfer coefficient on the MD design is evaluated. The design equations were explained, and a design algorithm was proposed. This algorithm indicated that the membrane thickness and velocity are the most important parameters that influence the transmembrane flux. While the influence of the membrane thickness was found to be direct, the influence of the velocity was indirect through the heat transfer coefficient.

Furthermore, according to our findings, it is difficult to identify the fitting heat transfer coefficient correlation for a theoretical design, which affects the transmembrane flux significantly. A flux discrepancy of up to 1 kg/m^2^/h is observed between different laminar-region heat-transfer coefficients. Therefore, it is vital to conduct a thorough experiment to identify the proper heat transfer correlation pertinent to the specific process in the design stage.

For low-grade heat effluents from factories such as breweries or bathroom wastes, it is recommended to hybridize the MD with NF as a pretreatment system, because of its ability to remove organic molecules from the waste.

## Figures and Tables

**Figure 1 membranes-12-01279-f001:**
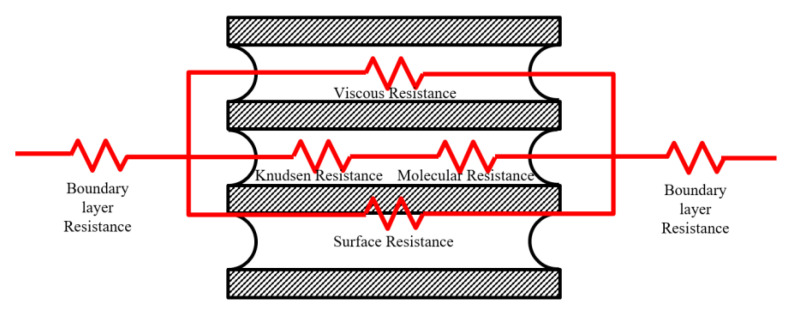
Serial and parallel arrangement of resistances to mass transport in MD [10].

**Figure 2 membranes-12-01279-f002:**
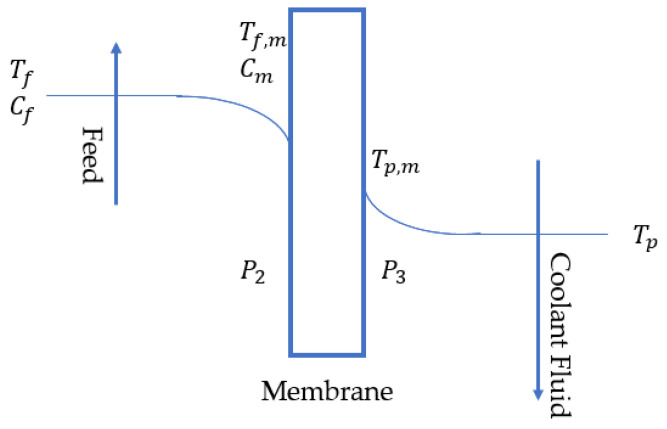
Temperature gradient in a direct contact membrane distillation system [12].

**Figure 3 membranes-12-01279-f003:**
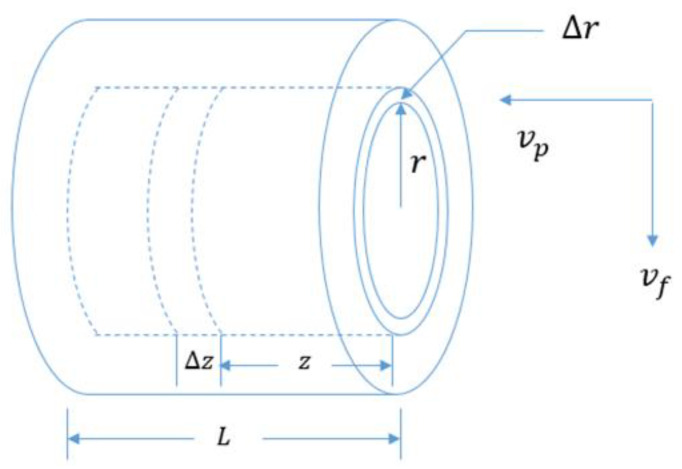
Differential material balance in a hollow fiber MD module with cross flow.

**Figure 4 membranes-12-01279-f004:**
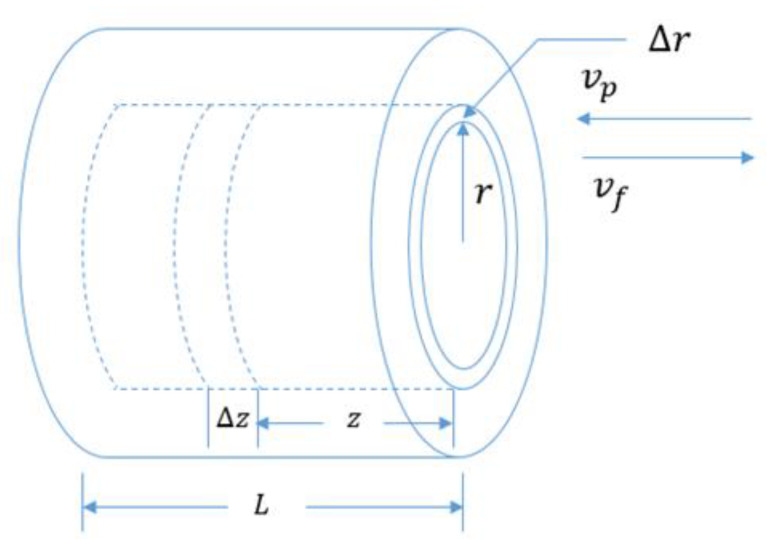
Differential material balance in a hollow fiber MD module with counter-current flow.

**Figure 5 membranes-12-01279-f005:**
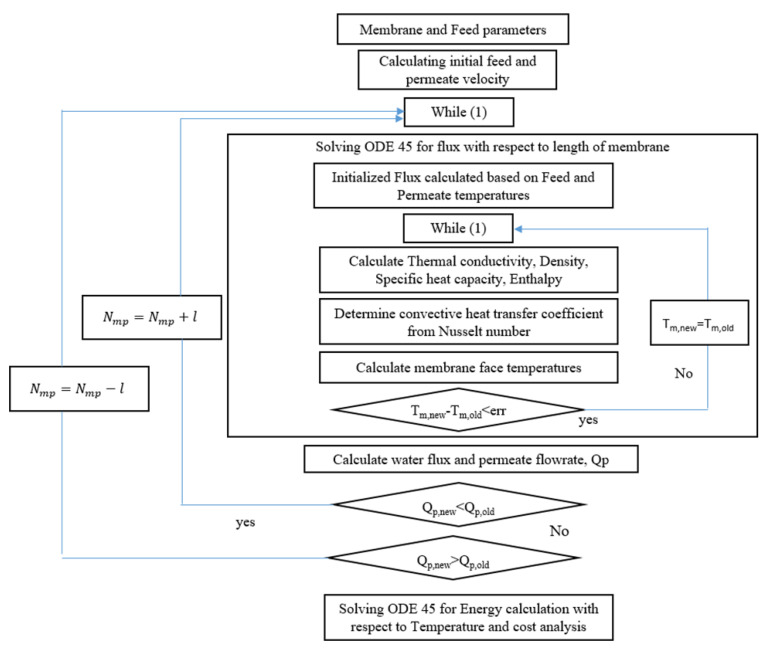
Design algorithm for the MD process.

**Figure 6 membranes-12-01279-f006:**
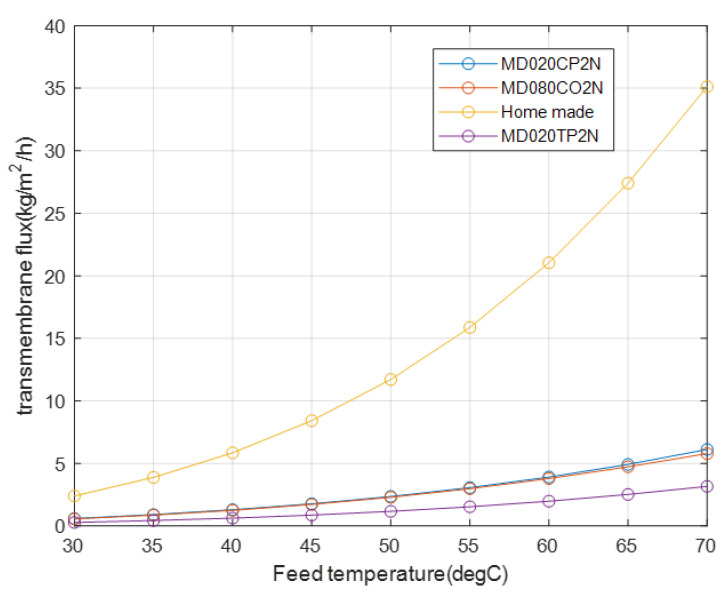
Transmembrane flux with respect to the feed temperature.

**Figure 7 membranes-12-01279-f007:**
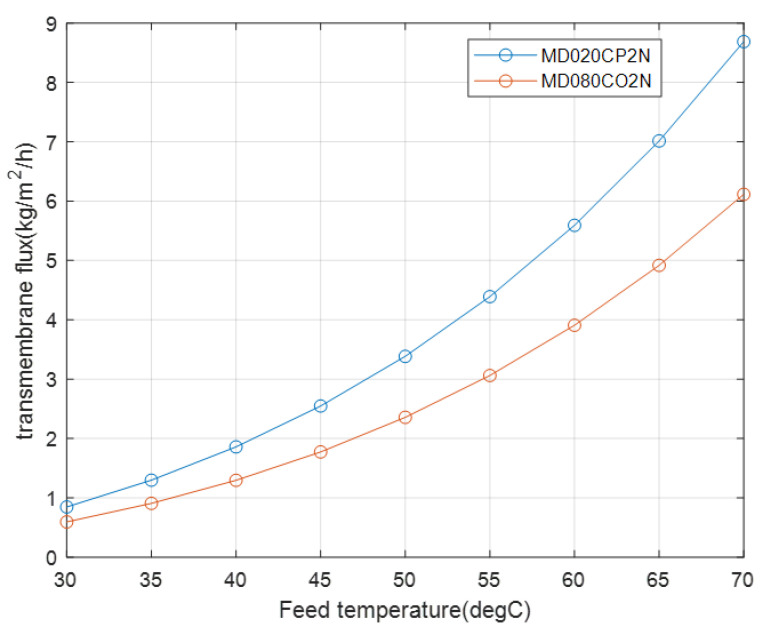
Transmembrane fluxes of MD020CP2N and MD080CO2N modules with different membrane thickness (0.05 and 0.65 mm) and the same number of fibers (40).

**Figure 8 membranes-12-01279-f008:**
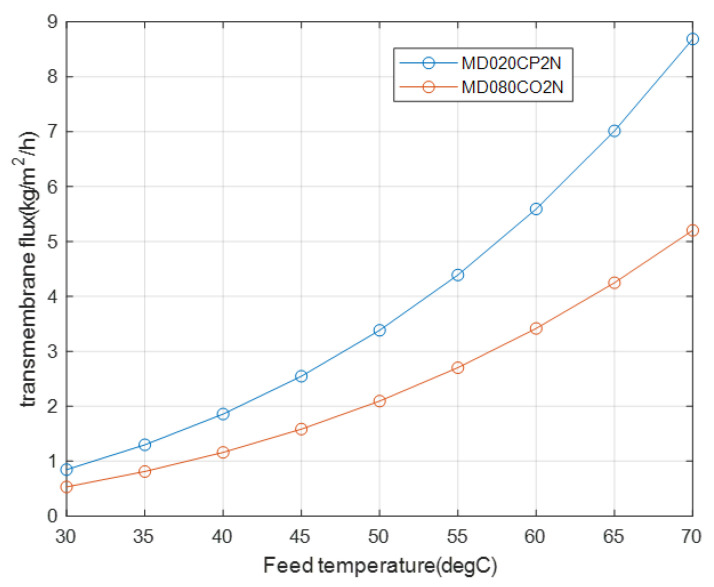
Transmembrane fluxes of MD020CP2N and MD080CO2N modules for the same membrane thickness (0.05 mm) and different number of fibers (40 and 467).

**Figure 9 membranes-12-01279-f009:**
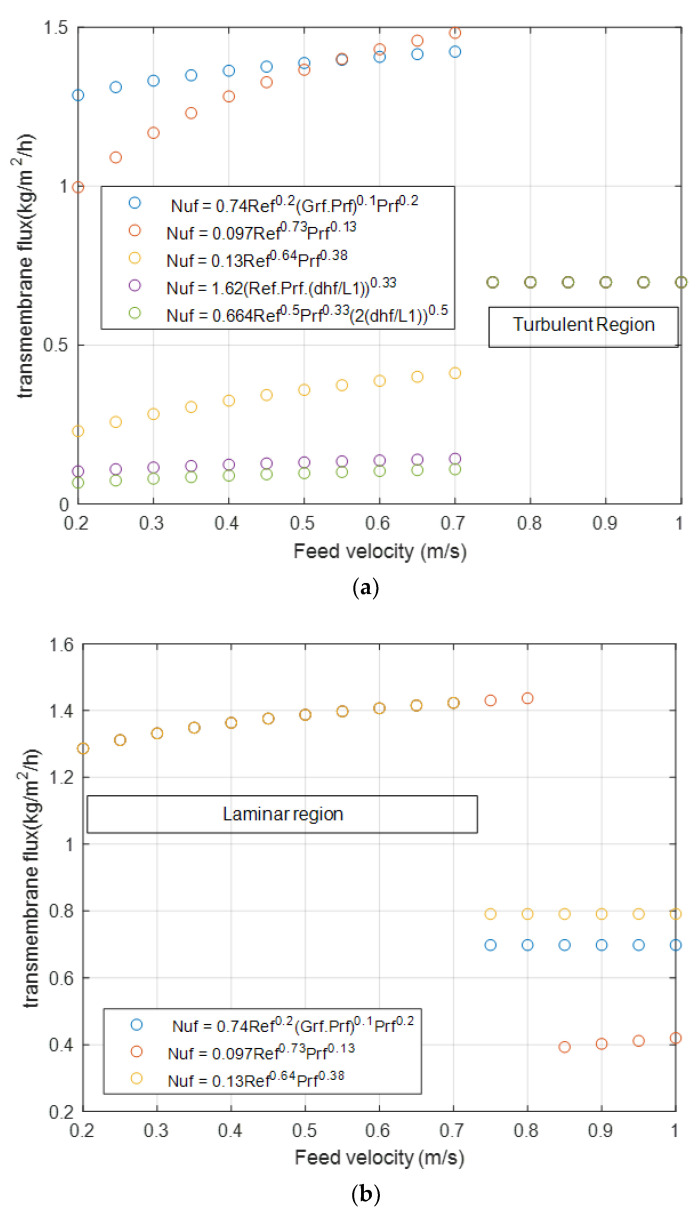
Transmembrane flux versus velocity for heat transfer coefficient correlations in the (**a**) laminar region and the (**b**) turbulent region.

**Table 1 membranes-12-01279-t001:** DCMD modules characteristics and membrane properties [11].

	MD020CP2N	MD080CO2N	Home-Made	MD020TP2N
Manufacturer	Microdyn	Enka-Microdyn	Home-made	Enka-Microdyn
Membrane material	Polypropylene	Polypropylene	Polypropylene	Polypropylene
Pore size (μm)	0.2	0.2	0.2	0.2
Fiber outer diameter (mm)	2.8	2.8	0.3	8.6
Fiber inner diameter (mm)	1.5	1.5	0.2	5.5
Membrane thickness (mm)	0.65	0.65	0.05	1.55
Number of fibers	40	467	1500	3
Shell inner diameter (m)	0.021	0.085	0.03	0.021
Packing factor	0.7	0.5	0.15	0.5
Length (m)	0.45	0.1	0.24	0.75
Surface area (m^2^)	0.1	2	0.35	0.036
Porosity	0.7	0.7	0.45	0.7
Tortuosity	1.4	1.4	2.2	1.4

**Table 2 membranes-12-01279-t002:** MD design parameters.

Design Parameters	Values
Number of fibers	3000
Fiber outer diameter (mm)	0.3
Fiber inner diameter (mm)	0.2
Membrane thickness (mm)	0.05
Shell inner diameter (m)	0.03
Packing factor	0.15
Length, m	1
Feed velocity, m/s	0.018
Permeate velocity, m/s	0.28
Feed-side temperature, °C	40
Permeate-side temperature, °C (the rest of the year except winter)	25
Permeate-side temperature, °C (winter period)	7

**Table 3 membranes-12-01279-t003:** MD cost estimation parameters.

Cost Parameters	Formula
Feed flow rate	$\frac{\pi }{4}v\left(1-\theta \right)\left[\left(D_{so}{}^{2}-D_{sii}{}^{2}\right)\right]N_{mp}$
Total membrane area	$A_{t}=\left(\frac{surface\;area}{volume}\right)*V_{m}*N_{mp}$
Plant availability, f	0.5
Plant capacity, w	$2400\;\left[m^{3}/\mathrm{day}\right]$
Plant life, n	20
Interest rate,i	0.05
$\mathrm{Pretreatment}\;\mathrm{cost},\;C_{pretreat}$	658(wRmd)0.8
$\mathrm{Membrane}\;\mathrm{cost},\;C_{membr}$	$90\;\left[\mathrm{USD}/m^{2}\right]$
$\mathrm{Total}\;\mathrm{cost}\;\mathrm{of}\;\mathrm{membrane},\;C_{tmemb}$	$C_{memb}A_{t}$
$\mathrm{Indirect}\;\mathrm{cost},\;\;C_{ic}$	$0.1*\left(C_{tmemb}+C_{pretreat}\right)$
$\mathrm{Annualized}\;\mathrm{capital}\;\mathrm{cost},\;C_{a}$	$C_{c}\frac{i{\left(1+i\right)}^{n}}{{\left(1+i\right)}^{n}-1}$
$\mathrm{Membrane}\;\mathrm{replacement}\;\mathrm{cost},\;C_{membr}$	$0.15\;*\;C_{tmemb}$
$\mathrm{Spare}\;\mathrm{cost},\;C_{sp}$	0.033*w*f*365
$\mathrm{Labor}\;\mathrm{cost},\;C_{lb}$	0.03*w*f*365
$\mathrm{Chemical}\;\mathrm{cost},\;C_{ch}$	0.018*w*f*365
$\mathrm{Waste}\;\mathrm{disposal}\;\mathrm{cost},\;C_{wd}$	0.0015*w*f*365
$\mathrm{Total}\;\mathrm{annual}\;\mathrm{cost},\;C_{ta}$	$C_{memba}+C_{membr}+C_{sp}+C_{lb}+C_{ch}+C_{wd}$
$\mathrm{Total}\;\mathrm{unit}\;\mathrm{product}\;\mathrm{cost},\;T_{upc}$	$C_{ta}/\left(W*f*365\right)$

**Table 4 membranes-12-01279-t004:** Results of the transmembrane flux parameters at 70 °C.

	$v_{f}$	$v_{p}$	$Re_{f}$	$Re_{p}$	$T_{f,m}$	$T_{p,m}$	ΔT	$C_{m}$
MD020CP2N	0.1926	0.2653	738.62	800	329.04	298.85	30.19	1.283 × 10^−7^
MD080CO2N	0.0165	0.0097	22.50	29.43	329.10	302.51	26.59	1.285 × 10^−7^
Homemade	0.2888	0.0466	20.21	20.98	336.41	317.02	22.39	6.883 × 10^−7^
MD020TP2N	0.1910	0.1607	1631.8	1889.3	332.94	300.63	32.31	5.3905 × 10^−8^

**Table 5 membranes-12-01279-t005:** Results of the transmembrane flux for different membrane thicknesses.

	$v_{f}$	$v_{p}$	$Re_{f}$	$Re_{p}$	$T_{f,m}$	$T_{p,m}$	ΔT	$C_{m}$
MD020CP2N	0.0594	0.2653	183.45	1170.5	316.67	313.31	3.36	1.669 × 10^−6^
MD080CO2N	0.1926	0.2653	738.62	800	329.04	298.85	30.19	1.283 × 10^−7^

**Table 6 membranes-12-01279-t006:** Transmembrane fluxes for different number of fibers.

	$v_{f}$	$v_{p}$	$Re_{f}$	$Re_{p}$	$T_{f,m}$	$T_{p,m}$	ΔT	$C_{m}$
MD020CP2N	0.0594	0.2653	183.45	1170.5	316.67	313.31	3.36	1.669 × 10^−6^
MD080CO2N	0.0051	0.0227	12.62	98.11	315.48	313.42	2.06	1.668 × 10^−6^

**Table 7 membranes-12-01279-t007:** MD design and cost results for the homemade module.

Cost and Design Parameters	Results Based on Feed Velocity		
	0.2	1	2.5
Production capacity [m^3^/h]	100	100	100
Number of modules in parallel, $N_{mp}$	2050	1652	6855
Total membrane area, $A_{t}\;\left[m^{2}\right]$	3.256 × 10^4^	2.624 × 10^4^	1.089 × 10^5^
Recovery, $R_{md}$	0.0297	0.0074	0.00071
Pretreatment cost [USD]	5.552 × 10^6^	1.693 × 10^7^	1.100 × 10^8^
Total annualized cost, $C_{ta}$ $\left[\frac{\mathrm{USD}}{\mathrm{yr}}\right]$	1.253 × 10^6^	2.122 × 10^6^	1.211 × 10^7^
Total unit product cost from low-grade wastes, $T_{upc}$ $\left[\frac{\mathrm{USD}}{m^{3}}\right]$	1.5897	2.6912	15.36
Total unit product cost from fresh cold seawater, $\left[\frac{\mathrm{USD}}{m^{3}}\right]$	38.84	52.20	83.32

## Data Availability

Not applicable.

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
