# Peer review of "Design Parameters of a Direct Contact Membrane Distillation and a Case Study of Its Applicability to Low-Grade Waste Energy"

_membranes, 2022, doi:10.3390/membranes12121279_

Round 1
Reviewer 1 Report
The work comprehensively reviewed the main physical and physio- chemical properties that affect MD process design. And then a MD design algorithm was developed and cost analysis of the design was also carried out. The work could give help for understanding MD process design, mass transfer and energy consumption mechanisms. While there are several key problems should be improved.
1. The English expression is good, while some miss typing still exists, as in line 723.
2. As we all know the thickness and feed velocity are key factors affecting the mass and heat transfer efficiency in MD process. When taken the home-made membrane to compare the effect of thickness on MD performance, how about the simultaneous effect of the small out/inner diameter and the feed velocity owing to the different diameter of the fibers. You know we can compare the effect of thickness on MD performance using membranes with same outer or inner diameter but different thickness. So we could get same feed velocity, only the thickness changed. Please give more detailed introduction for this.
3. The distribution of the fibers in MD modules is also key factor affecting the flowing of feed solution/permeate gas flow. How about in this work? Should also be introduced.
Reviewer 2 Report
The paper study design parameters of direct contact membrane distillation system. In this study, the physical and physiochemical properties that affect the design of MD are reviewed comprehensively, and based on the reviewed parameters, an MD design algorithm is developed. In addition, a cost analysis of the designed MD process for low-grade-energy fluids is conducted.
In addition to poor novelty, there are several issues in the manuscript that should be addressed. Some of them are provided for improving the manuscript:
1. The novelty of the work should be highlighted.
2. The introduction is too long. It should be revised and reduced.
3. All the data and results are based on the algorithm, thus it should be highlighted through the title and the body of the manuscript.
4. What is the difference between this work and the work in reference [11]?
5. In reference [11], the home made membrane did not give the maximum transmembrane flux as your model. could you explain?
6. some references discussed the thermal performance of the MD system and can be used to improve the quality of the manuscript:
"Thermal analysis evaluation of direct contact membrane distillation system." Case Studies in Thermal Engineering 13 (2019): 100377.
"Novel membrane suitable for membrane distillation: Effect of mixed nanofillers on the membrane performance." Key Engineering Materials. Vol. 801. Trans Tech Publications Ltd, 2019.
7. Please modify the error in line 723.
8. Each parameter in the equations should be identified.
Round 2
Reviewer 2 Report
This paper can be accepted now since the authors have clarified all the questions.